# *Galleria mellonella* (Greater Wax Moth) as a Reliable Animal Model to Study the Efficacy of Nanomaterials in Fighting Pathogens

**DOI:** 10.3390/nano15010067

**Published:** 2025-01-03

**Authors:** Stefania Villani, Matteo Calcagnile, Christian Demitri, Pietro Alifano

**Affiliations:** 1Department of Engineering for Innovation, University of Salento, Via Monteroni, 73100 Lecce, Italy; stefania.villani@unisalento.it; 2Department of Experimental Medicine, University of Salento, Via Monteroni, 73100 Lecce, Italy; christian.demitri@unisalento.it

**Keywords:** *Galleria mellonella*, nanoparticles, nanocomposites, multidrug-resistant microbes, toxicity, in vivo model

## Abstract

The spread of multidrug-resistant microbes has made it necessary and urgent to develop new strategies to deal with the infections they cause. Some of these are based on nanotechnology, which has revolutionized many fields in medicine. Evaluating the safety and efficacy of these new antimicrobial strategies requires testing in animal models before being tested in clinical trials. In this context, *Galleria mellonella* could represent a valid alternative to traditional mammalian and non-mammalian animal models, due to its low cost, ease of handling, and valuable biological properties to investigate host–pathogen interactions. The purpose of this review is to provide an updated overview of the literature concerning the use of *G. mellonella* larvae as an animal model to evaluate safety and efficacy of nanoparticles and nanomaterials, particularly, of those that are used or are under investigation to combat microbial pathogens.

## 1. Introduction

Nowadays, the continued and rapid emergence of resistant bacteria threatens the extraordinary health benefits that have been achieved with the discovery of antibiotics and their use in the treatment of infectious diseases. This threat is global, and its origin is attributed, on the one hand, to the indiscriminate or uncontrolled use of these drugs in various sectors [1], to environmental drivers of antibiotic resistance that are still poorly identified [2], and, for a long time, to the lack of antimicrobial drug development programs by pharmaceutical companies to address this challenge [3,4]. This has produced a worrying innovation gap in the treatment of infectious diseases.

In the meantime, many bacterial pathogens have emerged as public health threats, such as Carbapenem-resistant Enterobacteriaceae (CRE) [5], Methicillin-resistant *Staphylococcus aureus* (MRSA) [6], *Mycobacterium tuberculosis* [7], *Neisseria gonorrhoeae* [8], *Acinetobacter baumannii* [9], *Pseudomonas aeruginosa* [10], and *Salmonella* [11]. Many of these bacteria are difficult to treat because they are resistant to conventional antibiotics, including carbapenems, which are often used as a last resort to treat serious infections. Because of the resistance of these bacteria to conventional antibiotics, the World Health Organization (WHO) has identified a priority list that includes bacterial pathogens as high-priority targets for the research and development of new treatments and vaccines [7]. For these reasons, several innovative therapies have been developed to fight bacterial infections, such as bacteriophages [12], antibody-based treatments [13], anti-virulence therapies [14,15], and nanoparticles [16,17,18]. In addition, the development of antibacterial materials, which are non-toxic for humans, could help reduce the spread of nosocomial bacteria [19]. Developing innovative therapies and new antibiotics is a complex and challenging process for several reasons, including the high development costs [3,4]. This is partly due to the development process that can take many years and requires extensive preclinical and clinical testing to ensure safety and efficacy.

To reduce the cost of preclinical trials, it is essential to develop alternative animal models that can be easily managed in the laboratory and at a low cost. Mice are a standard animal model for many types of research, but, in recent years, there has been a growing interest in developing alternative animal models such as zebrafish and insects [20,21,22,23]. Insects, such as fruit flies (*Drosophila melanogaster*) [22], Coleoptera (*Tenebrio molitor*) [23], and Lepidopthera [21], have become important models for biomedical research due to their ease of handling, low cost, and fast reproduction rate and reduced ethical concerns compared to mammals. Insects are used to study infections and for drug screening to determine their efficacy and toxicity before moving on to more expensive animal models. Furthermore, insects can be used to study social interactions [24].

Combating emerging pathogens is a complex challenge, and innovative materials and nanotechnology may be future therapies. In this study, we focus on an innovative animal testing model, *Galleria mellonella,* to test methods to fight pathogens with nanotechnological devices. *G. mellonella* is a lepidopteran widely used for laboratory studies, especially for toxicological and pharmaceutical experiments. The first infection and toxicity tests using *G. mellonella* larvae as model organisms date back to the mid-1900s. However, the literature indicates that research on this insect began as early as the early 1900s, studying *Mycobacterium tuberculosis* infections [25,26]. *G. mellonella* larvae have been widely utilized to study pathogenic microorganisms, conduct pharmacological research—including toxicology and drug efficacy studies—and investigate the innate immune system [21,23,25,27,28,29].

The aim is to provide an overview of the immune system of this insect and to discuss the literature concerning innovative materials and nanotechnologies to combat bacterial pathogens using *G. mellonella* as an in vivo model. The recent growth of nanotechnologies has impacted various fields, from biomedicine to industrial applications, but the involvement of in vivo models is required to assess their safety and efficacy. In this context, *G. mellonella* could represent a valid alternative to traditional mammalian and non-mammalian animal models, especially because of its cheapness, ease of handling, and absence of adaptive immunity [30] *G. mellonella* not only allows the evaluation of in vivo toxicity of nanoparticles, polymers, and nanocomposites, but also their in vivo efficacy as new antimicrobial strategies when it is used as a model of infection.

## 2. *G. mellonella* as an Animal Model of Infection: Quantitative Data

*G. mellonella* (commonly known as the greater wax moth) is a species of moth that is native to Europe, Asia, and North Africa [31]. It is a widely studied insect model organism in biology and has been used as a model for many biological and biomedical studies, particularly those related to immunology, microbiology, and toxicology [27,32]. These larvae have been shown to have a very robust immune system, making them a useful model organism for studying the host–pathogen interactions of various bacteria [21,28] and fungi [33]. One of the advantages of using *G. mellonella* in research is their small size, which makes them easy to handle and maintain in a laboratory setting [34]. This insect also has a short life cycle, allowing for rapid experimentation and results. Additionally, the larvae of *G. mellonella* are inexpensive to rear, and are ethically acceptable for animal experimentation since they do not have the same level of consciousness as higher animals. *G. mellonella* larvae are increasingly used as infection models to test the pathogenicity of Gram-positive bacteria, Gram-negative bacteria, and pathogenic fungi.

A search was performed on the Web of Science (WOS) database using the keywords “*Galleria mellonella* infection” and including papers published between 2002 and 2024. A total of 2501 studies were found that used this animal model for research purposes. The *G. mellonella* model has been increasingly used over the years until more than 250 annual studies have been published in the last 3 years (2022–2024) (Figure 1A). Most of the papers have been published in journals that WOS classifies in the following categories (Figure 1B): microbiology (n° 1203; 48%); infectious diseases (n° 484, 19%); immunology (n° 398, 16%); pharmacology pharmacy (n° 297, 12%); zoology (n° 165; 7%); multidisciplinary sciences (n° 183, 7%); biochemistry and molecular biology (n° 175, 7%); entomology (n° 141; 6%); mycology (n° 150, 6%); biotechnology and applied microbiology (n° 144, 6%). Several documents were research articles and reviews that were published on this topic (Figure 1C). According to the WOS database, *Galleria mellonella* has been widely used to study various pathogens, including bacteria and fungi (Figure 1D). Among fungi, *Candida albicans* is the most studied, with 349 documents. For pathogenic bacteria, Gram-negative species are predominant, including *Pseudomonas aeruginosa* (358 documents), *Acinetobacter baumannii* (219 documents), and enterobacteria, with *Klebsiella pneumoniae* being the most studied (256 documents). Among Gram-positive bacteria, *Staphylococcus aureus* (117 documents) and *Streptococcus* spp. (88 documents) are the most studied. Expanding the search to include studies linking “*Galleria mellonella*” with materials or formulations used in nanotechnology, numerous studies focus on metals (111 documents), such as silver (70 documents), copper (53 documents), gold (34 documents), and oxides in general (72 documents) (Figure 1E). Additionally, nanoparticles are mentioned in 51 documents, chitosan in 22 documents, and the term “polymeric” in 20 documents (Figure 1E).

The already published reviews covered several topics including the study of pathogenic fungi [33] and pathogenic bacteria [35,36,37]. Some of these reviews deal with a specific pathogen or a very restricted set of bacteria. For instance, some of the documents deal with *A. baumannii* [36], *Klebsiella pneumoniae* [37], *Aspergillus fumigatus* [33], *Enterococcus* [35], and *P. aeruginosa* [38]. Other interesting papers also discuss practical aspects concerning *G. mellonella* rearing, larvae management in laboratory conditions, and methods of infection by inoculation [34,39,40]. Most of the recently published reviews offer an overview of the studies carried out with *G. mellonella* regarding the drug testing of antibacterial and antifungal chemical agents [21,25,28,30,41,42,43]. The advancement of nanotechnologies, however, has allowed the development of new antimicrobial strategies, like metal nanoparticles with antibacterial properties or smart nanomaterials for biomedical applications. In this context, the present review has the purpose of carrying out an in-depth and unedited analysis of the experiments carried out so far on *G. mellonella* as an in vivo model of nanotoxicity.

### 2.1. A General Overview of the Innate Immunity in Insects

Insects have an innate immune system that comprises two main types of immune responses: the humoral response and the cellular response. Some fundamental aspects of the insect innate immunity, which are relevant to the purpose of this review, are discussed below, while more in-depth literature reviews are already available [42,43].

The cellular component of the insect hemolymph is composed of different types of hemocytes [44,45] that are responsible for different immunological responses, including nodulation, encapsulation, and phagocytosis [42,46]. Nodulation and encapsulation serve the purpose of isolating pathogens, with nodulation directed against microorganisms and encapsulation against larger pathogens [42]. In both cases, myelination occurs, which is crucial for wound-healing and immune processes [29].

Humoral immune response includes the production of antimicrobial peptides (AMPs), the activation of prophenoloxidase, and the production of reactive oxygen species [47,48]. The phenoloxidase [49] plays a vital role in this process and is activated by certain receptors, such as peptidoglycan recognition proteins (PGRPs), that detect the presence of a bacterial pathogen. The activation of this signaling system generates a cascade of serine proteases that culminates in the activation of the enzyme that triggers melanin production [50,51].

AMPs are small peptides that can rapidly kill or neutralize invading microorganisms such as bacteria and fungi. AMPs are synthesized and secreted by body cells [47] and, to a lesser extent, by hemocytes and epithelial cells. AMPs have several mechanisms of action that allow them to kill or neutralize microorganisms. Some AMPs can bind to and disrupt the microbial cell membrane, causing the leakage of cellular contents and cell death [52,53,54,55]. Other AMPs can inhibit bacterial DNA and protein synthesis or interfere with other vital cellular processes [56,57]. Insects produce a diverse array of AMPs effective against different types of microorganisms. This diversity of AMPs allows insects to protect themselves from a wide range of pathogens. Insects can also modulate the expression of AMPs in response to microbial infection. Following an infection, AMP levels rise from undetectable concentrations to micromolar concentrations [58]. AMPs have been extensively studied in insects and have shown promise as potential therapeutic agents for human diseases [59,60]. Many of the AMPs produced by insects are non-toxic to mammalian cells [60], making them an attractive alternative to conventional antibiotics, which are associated with the emergence of antibiotic-resistant bacteria [61,62]. AMPs in insects can be divided into four families according to their structures or unique sequences. These include the α-helical peptides (cecropin and moricin), the cysteine-rich peptides (defensin and drosomycin), the proline-rich peptides (apidaecin, drosocin, and lebocin), the glycine-rich peptides/proteins (attacin and gloverin [59] and lysozyme). Insects have a closely interconnected immune system where the humoral and cellular immunity mechanisms work together in an effective cross-talk.

The activation of the insect immune response involves a signaling pathway that is triggered by the detection of microbial pathogens by specific receptors on immune cells. These receptors, called pattern recognition receptors (PRRs), can recognize conserved molecular patterns (pathogen-associated molecular patterns, PAMPs) [63] present in different classes of pathogens. Among these receptors are the immunolectins, PGRP, β-1,3-glucan recognizing proteins, hemolin, and integrins [48]. Some of these receptors are more responsive to a specific class of pathogens: for example, in *Drosophila*, the PGRP-SC1 receptor is implicated in the recognition of *S. aureus* [64], the GNBP1 and PGRP-SA receptors physically interact for the recognition of Gram-positive bacteria [65], and, finally, the membrane-bound PGRP-LC and secreted/cytosolic PGRP-LE receptors are implicated in the recognition of Gram-negative bacteria [66]. Upon the detection of a pathogen, the receptors activate downstream signaling pathways that involve various intracellular signaling molecules, including serine proteases, focal adhesion kinase (FAK), and mitogen-activated protein kinases (MAPKs) [48]. Insects, like mammals, have three pathways of immune response activation: the Toll pathway, the Immune Deficiency (IMD) pathway, and the Janus Kinase/Signal Transducer and Activator of Transcription (JAK-STAT) pathway [67,68].

A relevant aspect when *G. mellonella* is used as an infection model is the determination of the microorganism burden over time after the infection for immunological or toxicological investigations. For instance, *C. albicans* load was determined by colony counting by collecting the larval homogenate before and after the treatment with ribavirin as a monotherapy or in combination with caspofungin [69]. Alternatively, hemolymph bacterial burden can be determined by colony counting, as was performed for *P. aeruginosa* after the in vivo administration of cefotaxime, piperacillin, meropenem, amikacin, levofloxacin, and colistin [70], and for *S. aureus* to test the synergistic effect of linezolid and fosfomycin administration [71]. *G. mellonella* hemocytes are known to produce reactive oxygen species as a strategy to fight infections, and this immunological side can be exploited to monitor the bacterial load over time. This strategy has been used to determine *G. mellonella* bacterial burden over time after the infection with *Streptococcus pneumoniae*: at different time points, and larval hemolymph was collected and processed by using Electron Paramagnetic Resonance Spectroscopy (EPR), giving important information about the activity of *G. mellonella* hemocytes against bacterial infection [72]. The cytological analysis on larval hemocytes has also been useful for the in vivo evaluation of the antifungal activity of *Origanum majorana L.* essential oil on *Candida albicans*. The in vivo administration of *O. majorana* essential oil seems to induce morphological changes in hemocytes, increasing the actin polymerization with the formation of pseudopods, and stimulating *C. albicans* phagocytosis [73].

### 2.2. Monitoring Infection in the Context of the Immune System of G. mellonella

One of the advantages of using *G. mellonella* as an in vivo infection model is that several techniques can be used to investigate the mechanisms of the innate immune response at the molecular and cellular levels [74]. The methods that can be used include protocols for examining cellular and humoral immunity. This section first discusses some general methods for performing infection experiments and then the principal strategies by which the humoral immune response in *G. mellonella* larvae can be analyzed.

The experimental design of larval infection is of critical importance, and several aspects should be considered before experimenting. Among the most important factors to consider are the microbial charge of the inoculum and the incubation temperature, which can affect the health of the larvae [75,76]. On the other hand, observational methods or more complicated methods that give more significant indications can be used to evaluate the outcome of the infection experiment. The simplest method to evaluate an infection is the observation of the phenotype of the larvae, which, if healthy, appear cream-colored. When a larva is infected, its color changes to a light brown, which becomes darker until it turns almost black in dead larvae [39,77]. This dark phenotype is determined by the myelination process described above. The myelination can also be assessed using light absorption measurement. In this case, the lymph must be extracted, and a buffer must be added, as the myelination and the coagulation processes also occur spontaneously in the extracted lymph [77]. In addition, the competitive inhibitor (phenylthiourea) of the phenoloxidase enzyme can be used [78]. Some authors have used more complicated observational methods by mechanically stimulating the larvae and assessing the response (no movement; minimal movement on stimulation; movement when stimulated; movement without stimulation) [79]. It is also possible to evaluate how many larvae manage to form the cocoon [79]. These methods have the advantage of being simple and requiring standard tools. On the other hand, these methods do not allow the molecular mechanisms associated with infection to be investigated.

Six prevalent types of hemocytes have been reported in *G. mellonella* hemolymph: prohemocytes, which are small, rounded cells that can divide and differentiate into other cell types; plasmatocytes and granulocytes, which are the predominant cell types and participate in phagocytosis, nodule formation, and encapsulation; spherulocytes, which show a variable number of small spherical inclusions; oenocytes, which are large, binucleate and non-phagocytic cells that contain prophenoloxidase; coagulocytes, which are involved in the clotting process [27,80]. Some of the methods used to study hemocytes as indicators of toxicity are further described below.

As previously mentioned, humoral immunity includes the production of AMPs, opsonins, and melanin. Among the identified or presumed AMPs are two cercopins, galiomycin, gallerimycin, gloverin, lysozyme, a helix-like peptide, an inducible inhibitor of serine protease 2, two anionic Gm peptides, two proline-rich Gm peptides 1 and 2, five types of moricin-like peptides, and X-tox [27]. As previously mentioned, the levels of AMPs can rise from undetectable concentrations to micromolar concentrations following an infection [58]. For this reason, these AMPs can be used as indicators of infection by quantification by molecular methods, such as RT-qPCR and proteomics, or by in vitro antibacterial assays (agar diffusion assay, minimum inhibitory concentration (MIC), etc.) [81].

The method based on agar diffusion is simple and is used to evaluate the bactericidal activity of bacterial isolates, single molecules, or nanoparticles [82,83,84]. This method allows the direct evaluation of the bactericidal activity of hemocyte-free hemolymph or previously isolated AMPs [85,86,87]. To carry out this experiment, the hemolymph or the solubilized peptide is inoculated in the center of a Petri dish containing an agar growth medium onto which an indicator microorganism has been seeded. The indicator microorganism can be a reference strain or a specific strain on which the antibacterial effect is evaluated. Furthermore, some microorganisms are sensitive to the action of an antibacterial enzyme. For example, *Micrococcus luteus* is commonly used to evaluate lysozyme activity [86]. However, other AMPs also could have bactericidal activity against *M. luteus* [88]. Hence, this method can be employed as a semi-quantitative method to assess the presence of AMPs but has low specificity to differentiate these peptides. The best methods to quantitatively and qualitatively evaluate AMP production are proteomics and transcriptomics. The hemolymph treated to eliminate lipids and deprived of the cellular component can be subjected to chromatographic methods. Sodium dodecyl sulfate polyacrylamide gel electrophoresis (SDS-PAGE) is a method that separates all hemolymph proteins into bands based on their molecular weight. The resulting gel can then be used to perform the bioautography technique, which allows us to determine which bands in the gel are AMPs. To achieve this method, the gel is treated to renature the proteins and is overlaid with a layer of solid culture medium containing the bacteria (overlay) [89,90,91].

Transcriptome analysis (RNA-seq) can be used to assess the response of *G. mellonella* to infection [89,92,93]. This method is employed to quantify the transcript level of immune system genes as well as other lepidopteran genes. However, this method is expensive. For this reason, an analysis of the transcription levels of AMP-specific genes is often preferred. Various studies use these methods to assess the transcription of AMPs during the infection of fungi and bacteria [89]. This method also effectively differentiates intestinal symbionts from pathogenic microorganisms [93]. RNA-seq analysis and RT-qPCR can also be used to analyze genes inherent to the host–pathogen interaction of microorganisms used to infect larvae.

The methods listed above can be used to qualitatively and quantitatively analyze other proteins and genes related to the immune response, such as ApoLp-III, PGRPs, and *Galleria mellonella* collagen protein 8 (GmCP8). *G. mellonella* produces multiple opsonins that can recognize and bind to conserved microbial components, such as lipopolysaccharides (LPSs), lipoteichoic acid (LTA), and peptidoglycan. ApoLp-III acts as an opsonin: it binds to the surface of pathogens and acts as a bridging molecule to facilitate their recognition and uptake by hemocytes. It has been shown to bind to bacterial LPSs [94], LTA [95], and fungal β-glucan [96] and has been implicated in the recognition of several pathogenic bacteria and fungi in *G. mellonella* [97]. GmCP8 is a protein that has been shown to bind to several bacterial and fungal species, including *Pseudomonas entomophila*, *P. aeruginosa*, *Bacillus thuringiensis*, *S. aureus*, *Escherichia coli,* and *C. albicans* [98]. In fact, this protein binds several ligands such as LPSs, LTA, and β-1,3-glucan [99].

## 3. *G. mellonella* Larvae as an Animal Model for Nanotoxicity Studies

The term nanoscience refers to the investigation of molecules and structures whose sizes are between 1 and 100 nm, involving physics, material science, and chemistry. The use of the prefix “nano” is derived from Greek, and it refers to the billionth part of a meter (10^−9^ m). Nanotechnology, however, refers to the transfer of theoretical studies to potential applications through observations, measurements, manipulations, and the assembly of the matter at the nanoscale [100]. In 1959, during the annual meeting of the American Physical Society, the physicist and Nobel Prize winner Richard Feynman introduced the concept of nanotechnology, giving a speech entitled “There’s Plenty of Room at the Bottom” at the Californian Institute of Technology (Caltech). Feynman formulated the hypothesis of the construction of smaller machines down to the molecular level; as a result, today, Feynman is considered the father of modern nanotechnologies [101]. In 1974, the Japanese scientist Norio Taniguchi introduced the term “nanotechnology” defining it as the process of separation, consolidation, and deformation of materials by one atom or one molecule [102]. The definition of nanomaterial was introduced by the European Commission (EC) in 2011, about natural, incidental, and manufactured materials and considering size as the most significative parameter of classification. To be considered as such, a nanomaterial has to possess 50% or more of the constituent particles with one or more external dimensions in the size range of 1–100 nm or a volume-specific surface area larger than 60 m^2^/cm^3^ [103]. According to this classification, nanoparticles (NPs) can be defined as nanomaterials that possess three dimensions in the nanoscale range [104].

In the early XXI century, interest in nanoscience and nanotechnologies grew, and numerous studies highlighted the great potential of matter at the nanoscale in various fields, from electronics to engineering, up to biology and medicine. Several studies describe the progress of nanotechnologies for biomedical applications like drug delivery and tissue regeneration, in addition to their great application for engineering and science material applications [105]. However, the fields of drug discovery, toxicology, and host–pathogen interactions need to select animal models for in vivo investigations. The choice of the murine model is generally preferred because of its relatively high similarity to human metabolism and immune system. However, the use of this mammalian model is expensive and painstaking and creates inquisitiveness due to a number of ethical restrictions [41,106]. For the same reasons, the involvement of non-mammalian animal models has been recently promoted, like the nematode *Caenorhabditis elegans* [107], the fruit fly *D. melanogaster* [108], the zebrafish *Danio rerio* [109], marine mussels like *Mytilus galloprovincialis* [110], and *Brachidontes pharaonis* [111] and, more recently, the larvae of the wax moth *G. mellonella*. *G. mellonella* larvae are not expensive and easy to be manipulated without ethical restrictions, allowing many replicates to be performed for more valid results.

As previously mentioned, the toxicity of nanomaterials and nanoparticles can be assessed using other experimental models, including mammals (mainly mice), other invertebrates, or in vitro cellular models. As for cellular models, different types of human or mammalian cell lines can be used. For example, the HeLa cell line is used because of its ease of use [112]. Other suitable cell lines include Calu-3 for respiratory epithelial models [113] and Caco-2 as an intestinal epithelial model [114]. More recently, the optimization of a more complex 3D-tetraculture system has allowed the evaluation of the sensitivity of nanomaterials mimicking the alveolar barrier [115]. The main methods to assess toxicity include lactate dehydrogenase (LDH) assays and MTT assays [116]. Other methods include studying oxidative stress and investigating programmed cell death pathways, such as apoptosis. The efficacy of these models can be assessed using techniques like Western blotting to evaluate the expression of biomarkers related to infection or assays for quantifying cytokines associated with inflammation [117]. The main disadvantages of these cellular models include the high costs associated with laboratory instruments and consumables, and the need for specialized personnel. However, these models can provide results that closely mimic human infections. Additionally, in vitro cell models are versatile and can be used independently or in combination with *G. mellonella* models.

Several animal species, including pigs [118,119], rabbits [120,121], and ferrets (Basu et al., 2024), have been successfully used as models to study infections. Non-mammalian models, such as those used in aquaculture, have also been instrumental in studying infections in livestock [122]. However, there is limited information on the in vivo effects of nanoparticles and nanomaterials in mammals, primarily due to ethical constraints on animal use. In contrast, non-mammalian models, particularly fish, have proven effective for nanotoxicity studies. Mice remain the better model for studying the toxicity and efficacy of materials and nanoparticles. Mice are gold-standard models for studying different type of infections, such as intestinal and pulmonary infections [123,124]. In addition, gnotobiotic mice are associated with known microbial communities, providing useful information on the interaction between microbiota and pathogens [125]. The advantages of using this model include the availability of genetically modified animals, such as knockout or transgenic mice, the similarities of their immune system to that of humans (including the adaptive immune response), and the ability to raise mice in controlled environments, which helps limit variables that could influence infection outcomes. However, there are also disadvantages, such as many of the mouse strains used are derived from inbreeding, which reduces genetic variability and increases the risk of genetic diseases. The main advantages and disadvantages of using mammalian models compared to *G. mellonella* are reported in Figure 2.

Both mammals and *G. mellonella* have been successfully used to study the biodistribution of nanoparticles [126,127]. However, due to anatomical differences between humans and insects, mammals—particularly mice—are generally considered more suitable models for such studies. Despite this, *G. mellonella* offers significant advantages for rapid screening, reducing animal use, and providing a more ethically and economically sustainable approach to experimentation. Also, the correlation between *G. mellonella* larvae and mammalian models’ in vivo toxicity has been demonstrated. The Global Harmonising System of Classification and Labelling of Chemicals (GHS) classifies chemicals according to their physical, toxicological, and environmental hazards, leading to the development of safety data sheets (SDSs) that represent an international system for hazard communication [128]. A recent study measured the toxicity of 19 chemicals in vitro, in 3T3 and NHK cell lines, and in vivo using *G. mellonella* larvae, demonstrating that GHS category 5 chemical toxicity was better predicted using larvae than cell lines [129]. Another highlight in the application of *G. mellonella* larvae for toxicity studies is the absence of adaptive immunity, which can benefit scientific research to study in detail the innate immune response without interference [30].

### Nanotoxicity Assessment in G. mellonella Larvae

Several strategies are available to evaluate the toxicity of chemicals, drugs, and nanomaterials in vivo using *G. mellonella* larvae, as summarized in Figure 3. The first more rapid evaluation is the analysis of larval mortality: larvae are considered dead when there is no movement after an external stimulus [72,130]. Then, *G. mellonella* survival curves are generally plotted using the Kaplan–Meier method, a statistical approach that is commonly used in medical research to estimate the fraction of surviving subjects during a certain period of time after treatment [72,130,131]. This type of statistical tool is essential to study the survival fraction of a study group.

Together with the determination of the lethal dose (LD) and the lethal concentration (LC) of the investigated agent, it is possible to easily collect large amounts of larvae hemolymph for further analysis. The similarities between the cellular immune systems of *G. mellonella* and mammals allow the use of hemocyte density as an indicator of toxicity, as this was achieved in a recent study evaluating the toxicity of silver NPs (Ag NPs), gold NPs (Au NPs), and selenium NPs (Se NPs) by extracting the hemolymph and measuring the hemocyte proliferation [132]. The traditional method to determine hemocyte viability is by using a dye exclusion test based on trypan blue, because it is cheaper if compared with other approaches, like fluorescence dyes, antibodies, ELISA plate readers, or flow cytometry [133]. The determination of the number of dead cells by optical microscopy is highly accurate but also time-consuming. In this regard, the hemocyte count by using an automated cell counter was proposed and compared with optical microscopy analysis as a part of a toxicity study of zinc oxide NPs (ZnO NPs) [134].

In the context of the evaluation of the toxicity in vivo, the impact on *G. mellonella* oxidative metabolism and NP biodistribution has been considered. The activity of some antioxidant enzymes involved in the metabolism of the reactive oxygen species (ROS), like superoxide dismutase (SOD), catalase (CAT), glutathione peroxidase (GPx), glutathione-S-transferase (GST), and acetylcholinesterase (AChE), has been estimated after the injection of titanium oxide NPs (TiO_2_ NPs) [135], copper oxide NPs (CuO NPs) [136], and metals like copper and zinc [137,138]. The degree of toxicity of metal NPs is also influenced by their distribution throughout body organs and tissues of *G. mellonella* larvae. In this regard, Ag NP, Au NP, and Se NP bioaccumulation was estimated through a histological characterization [132], while CuO NP [136] and TiO_2_ NP [135] biodistribution was evaluated by determining the metal levels in tissues using an Atomic Absorption Spectrophotometer.

## 4. *G. mellonella* Model to Investigate Antimicrobial Properties of Nanoparticles, Their Toxicity, and the Underlying Mechanisms

The emergence of antibiotic-resistant bacterial strains leads to the need to develop new antibacterial strategies. In this regard, nanotechnologies can contribute through the development of nanomaterials, such as metal NPs, as non-traditional antibacterial agents against Gram-positive and Gram-negative bacteria [139]. Several mechanisms have been proposed to explain the antimicrobial activity of metal NPs, i.e., physical damage, ion leaching, ion dissolution, and the production of ROS, leading to the loss of the cell membrane integrity and interfering with some essential metabolic pathways [140]. Among metal NPs, Ag NPs are the most used for biomedical applications due to the strong antimicrobial activity of silver alone or in combination with other antimicrobial agents; as a result, the development of nanotechnologies leads to the Ag NPs’ incorporation in creams [141], wound-healing dressing [142], food packaging [143], and for catalysis [144]. Furthermore, the emergence of multidrug-resistant microorganisms around the world needs the development of new effective and safe strategies [16].

In this context, Ag NPs could represent a valid alternative to traditional antibiotics in the treatment of multi-drug-resistant microorganisms, in the disinfection of surgical instruments, and in fighting infections [16,17]. Ag NPs can penetrate through the cell wall and accumulate at the level of the inner membrane, causing mechanical damage [145], increased cell permeability, and the loss of cytosolic contents, resulting in cell death [17,146]. The interaction with the inner membrane could alter phosphate ions’ uptake and release, compromising the respiratory chain and, as a consequence, the cellular energetic metabolism [147]. Also, the release of metallic ions (Ag^+^) contributes to the antibacterial action of Ag NPs, causing an increased oxidative stress related to the generation of ROS and free oxygen radicals [147]. The Ag^+^ affinity for protein thiol groups could inhibit the cellular detoxification, leading to ROS accumulation and DNA damage [17,147]. A vast number of studies investigated the antibacterial properties of Ag NPs obtained with conventional and green synthesis approaches against Gram-negative bacteria like *E. coli* [148,149,150], *P. aeruginosa* [150,151,152], and *A. baumannii* [149] and against Gram-positive bacteria such as *S. aureus* [148,149,150,153] and *B. subtilis* [149,150]. For a more complete investigation, *G. mellonella* larvae were chosen as a model to evaluate the Ag NP toxicity in vivo (Table 1), and the mechanisms of action of nanomaterials used as antimicrobial agents in *G. mellonella*-infected larvae are summarized in Figure 4.

A recent study analyzed the toxicity of Ag NPs synthesized using a water kefir liquor to be used against *P. aeruginosa* infections, observing that 80% of the larvae infected with the bacterium and prophylactically treated with nanoparticles survived [154]. Focusing on *P. aeruginosa* infection treatments, green Ag NPs synthetized with *Citrus latifolia tan* leaf extract and with the fungus *Aspergillus flavus* have been studied to determine the antibacterial potential tested against different clinical strains. They not only showed killing abilities due to severe bacterial cell damage at the level of the membranes, DNA, and oxidative metabolism, but also these green Ag NPs are non-toxic for *G. mellonella*, so they may be potentially utilized in future biotechnological applications [155]. Also, the toxicity of Ag NPs synthesized by using the fungus *Aspergillus tubingensis* has been evaluated to assess their safety for subsequent studies on mammalian cells: this in vivo evaluation allowed to ascertain their safety and their potential use in biomedical applications [156]. A similar approach has been used to assess the toxicity of biogenic Ag NPs that have been obtained through the fungus *Bionectria ochroleuca*: once their safety was determined using *G. mellonella* as an in vivo model, green Ag NPs have been incorporated into cotton and polyesters, obtaining fabrics that showed strong antibacterial properties against *S. aureus, E. coli, C. albicans*, *Candida glabrata,* and *Candida parapsilosis* and interesting biofilm inhibition activity against *P. aeruginosa* [157]. Ag NPs could also be incorporated in polymeric matrices, obtaining a nanocomposite with interesting properties. For instance, the toxicity of Ag NP–chitosan nanocomposite has been evaluated in vivo using *G. mellonella* larvae: the low toxicity of this nanomaterial suggested its potential use in vivo as a topical treatment against murine cutaneous candidiasis [158]. Alternatively, it is possible to selectively deliver metallic nanomaterials, like silver nanoclusters (Ag NCs), by exploiting species-specific DNA aptamers against *P. aeruginosa* [159]: the targeting DNA-scaffold Ag NCs showed fast-acting antimicrobial activity in vitro and in vivo using the *G. mellonella* model.

In view of the possible biomedical applications, the evaluation of Ag NPs’ affinity for human blood proteins is necessary [160,161]. Recently, it has been demonstrated that Ag NPs could interact with albumin and α-1-acid glycoprotein in a way that is strongly influenced by NPs’ size, shape, and surface functionalization [162]. Similarly, the affinity of Ag NPs and Au NPs for human fibrinogen, transferrin, albumin, and apolipoprotein-A has been evaluated by Surface Plasmon Resonance (SPR), highlighting a different in vivo behavior [163]. To modulate the protein affinity of Ag NPs, they have been complexed with graphite oxide (GO), and the resulting nanocomposite material has been characterized by estimating the affinity for human proteins and the in vivo toxicity involving *G. mellonella* larvae [164]. Among inorganic NPs, rosehip extract-functionalized magnesium hydroxide NPs (Mg(OH)_2_ NPs) have been also investigated, exhibiting an increased antibacterial activity against *S. epidermidis*, *S. aureus,* and *E. coli* compared with those synthesized without functionalization, and they were also efficacious if administered in *S. aureus*-infected *G. mellonella* larvae [165].

The possibility to manage dimensions, shapes, and porosities leads to the application of metal oxide NPs (MO NPs) in biomedical and health-related industry [166]. Currently, several types of MO NPs are used in the biomedical field because of their antibacterial properties as wound-healing dressing, biosensors, and anticancer and image contrast agents [167]. Some examples are CuO NPs, ZnO NPs, iron oxide NPs (Fe_2_O_3_ NPs), silver oxide NPs (AgO NPs), MgO NPs, and TiO_2_ NPs [166]. The antibacterial activity of MO NPs is strongly influenced by some physico-chemical parameters like dimensions, shapes, charges, surface properties, and all other parameters that are affected by the synthesis protocol [168]. The main mechanism of action of MO NPs that has been suggested is the alteration of the oxidative metabolism through the ROS production, together with their dissolution, and the release of toxic free metal ions [169].

Despite the potential uses in the biomedical field, the toxicity of these nanomaterials and the possible side effects still need to be established. In this regard, the toxicity of some typologies of MO NPs has been evaluated by exploiting *G. mellonella* larvae as an in vivo model. CuO NPs, for example, are widely used in the biomedical field because of their antimicrobial properties against different Gram-positive and Gram-negative bacteria and for their fungicidal activity [170,171,172]. By force feeding *G. mellonella* larvae with four different CuO NP concentrations, the toxicity was analyzed by determining the LC_50_ and LC_90_, by the total hemocyte counting (THC) and by analyzing viable, apoptotic, necrotic, and micronucleated hemocytes. While the THC did not change significatively at all concentrations, the ratio of viable hemocytes decreased, and mitotic and micronucleated hemocytes increased with the increasing CuO NP concentration [173]. A similar study evaluated the metabolic impact of CuO NPs administered by force feeding *G. mellonella* larvae. A significative alteration has been shown in the activity of several enzymes involved in the adaptation to oxidative stress, like aspartate aminotransferase (AST), alanine aminotransferase (ALT), and lactate dehydrogenase (LDH), suggesting that the level of cell damage is strongly influenced by the exposure to CuO NPs [174].

In the in vivo evaluation of MO NP toxicity, it is essential to establish the effects of CuO NPs on tissue accumulation, as was carried out in a study that observed the NP accumulation in *G. mellonella* midgut and fat body tissues, countered by the increased antioxidant enzyme activity [136]. The effect of MO NPs on bioaccumulation has been also evaluated for TiO_2_ NPs, and it has been observed that most of the metal was eliminated through the Malpighian tubules. Also, it has been observed that the in vivo administration of TiO_2_ NPs causes oxidative stress and immune system alterations [135]. An already-mentioned study has focused on another typology of MO NPs that are ZnO NPs, which are in vivo administered by force feeding. To evaluate the toxicity of this nanomaterial, its lethal concentration was determined, and also the THC allowed the evaluation of NP toxicity in vivo [134]. ZnO NPs have also been exploited for their antifungal activity against *C. albicans*. The sub-lethal doses of ZnO NPs were chosen for a prophylactic injection in *G. mellonella* larvae followed by *C. albicans* infection. These results not only showed that ZnO NPs prolonged the survival of *C. albicans*-infected larvae, with a reduction in the fungal burden in the infected larvae, but also that these NPs seem to potentiate the host immunity system, exerting a protective effect [175].

Organic NPs, like liposomes, micelles, and polymeric nanocapsules, could be a valid alternative to the use of inorganic NPs for drug protection, diagnosis, and therapeutic purposes. As a result, the toxicity of lipid-core nanocapsules, coated with polysorbate 80, lecithin, and chitosan, has been evaluated in vivo by using *G. mellonella* animal model, demonstrating their safety and also the possibility to use larvae as a promising alternative in vivo model [176].

**Table 1 nanomaterials-15-00067-t001:** The evaluation of the toxicity and the antimicrobial efficacy of NPs in vivo using *G. mellonella*.

	Type	Synthesis Method	NPs’ Average Size (nm)	In Vivo Infection	Toxicity Assessment	Ref.
Metal NPs	Ag	Green| water kefir liquor	20	*P. aeruginosa*	Mortality–THC–bacterial load–phenoloxidase activity–nodulation assay	[154]
		Green|*C. latifolia tan* Biological|*A. flavus*	20	- *	Mortality	[155]
		Biological| *A. tubingensis*	35	- *	Mortality	[156]
		Biological|*B. ochroleuca*	8–21	- *	Mortality	[157]
	Ag–chitosan	Chemical	11	- *	Mortality	[158]
	Aptamer–AgNCs	Chemical	Unspecified	*P. aeruginosa*	Mortality	[159]
Inorganic NPs	Graphite oxide–Ag	Chemical	Unspecified	*S. aureus*	Mortality	[164]
	Mg(OH)_2_	Green|rosehip extract	90	*S. aureus*	Mortality	[165]
Metal Oxide NPs	CuO	Commercial|Nanokar	38	- *	LC_50_ and LC_90_–THC– hemocyte viability	[173]
		Commercial| Sigma-Aldrich	<50	- *	LC_10_ and LC_30_—metabolic and biochemical parameters–THC	[136]
		Commercial| Sigma-Aldrich	<50	- *	Bioaccumulation–metabolic enzyme activity	[174]
	TiO_2_	Commercial| Degussa P25	29	- *	Bioaccumulation–metabolic enzymes activity–total protein dosage–THC	[135]
	ZnO	Commercial| Alfa Aesar	70	- *	LC–THC	[134]
		Commercial| Sigma-Aldrich	100 × 15	*C. albicans*	Mortality–histopathological analysis–THC–bacterial load—SEM-phenoloxidase assay—Phagocytosis assays	[175]
Polymeric NPs	Lipid-core Nanocapsules	Chemical	150–190	- *	Mortality	[176]

* The symbol indicates that *G. mellonella* has been used as an in vivo model for toxicity studies, without in vivo infection. NPs: nanoparticles; Ag: silver; NCs: nanoclusters; Mg(OH)_2_: magnesium hydroxide; CuO: copper oxide; TiO_2_: titanium oxide; ZnO: zinc oxide; THC: total hemocyte counting; LC: lethal concentration; SEM: scanning electron microscopy.

### In Vivo Administration of Functionalized Nanoparticles for Toxicity Modulation and Drug Delivery

Together with the possibility to use *G. mellonella* larvae for nanoparticles’ toxicology studies, its involvement in drug testing is also discussed (Table 2), passing from antimicrobial and antifungal drug testing [21,30] to medicinal plants [177], food preservative agents [178], and drug nanoformulations for biomedical applications [179]. Often, inorganic and polymeric NPs are used as drug delivery carriers, and *G. mellonella* is generally preferred to test their toxicity and efficacy. For instance, Ag NPs could be exploited as medical agents for antimicrobial photodynamic therapy: to obtain a better performance, they have been conjugated with a phenothiazinium photosensitizer such as methylene blue and its derivatives, and the resulting toxicity has been evaluated in vivo through larval administration [180].

Otherwise, Ag NPs have been conjugated with cinnamaldehyde, the major chemical constituent of cinnamon essential oil reported from *Cinnamomum* spp., obtaining entrapped Ag NPs whose antibacterial efficacy has been evaluated in the *G. mellonella* larval model against multidrug-resistant strains of enteroaggregative *E. coli* (EAEC) [181]. Further in vivo studies using *G. mellonella* larvae analyzed the impact of biomolecule coronas on the nanotoxicity of metal NPs, like Ag, CuO, and ZnO NPs, showing that their antifungal activity was significantly reduced due to the acquisition of pathobiological or ecological biomolecule coronas by impairing the NPs’ bond to fungal spores [182].

In the development of effective antibacterial treatments against multidrug-resistant bacteria, Au nanostars have been tested alone or in association with amikacin to eradicate carbapenem-resistant *K. pneumoniae* biofilms, and this treatment was found to be tolerated if administered in vivo to *G. mellonella* larvae [183]. Among inorganic NPs, lycopene-impregnated mesoporous silica nanoparticles have been tested as a possible alternative treatment against vulvovaginal candidiasis. Although the antifungal results were not encouraging, in vivo experiments conducted in *G. mellonella* have allowed to ascertain their safety, and this aspect makes them further applicable in drug delivery [184].

Among drug carriers, polymeric NPs stand out for the delivery of antifungal, antibacterial, anticancer, and wound-healing drug delivery. Alginate, as a natural, non-toxic, biocompatible, and non-immunogenic polymer, has been chosen for the synthesis of NPs to deliver miltefosine in the treatment of *C. albicans*, *Cryptococcys neoformans* and *Cryptococcus gattii* [185], and *Candida auris* [186] infections. Lipid-based NPs have been involved in the drug delivery of antifungal agents, like itraconazole for the treatment of *Sporothrix brasiliensis* and *C. albicans* skin infections [187], anidulafungin against *C. albicans* [188], and *Lippia sidoides* essential oils against *C. auris* [189], optimizing cutaneous localization without compromising the efficacy. Lastly, poly-(lactic-co-glycolic) acid (PLGA) NPs conjugated with coumarin 6 and pterostilbene have been tested for their antifungal activity against *Aspergillus brasiliensis,* the causative agent of otomycosis in the external auditory canal [190].

In this context, the *G. mellonella* model could be useful to evaluate the biocompatibility and efficacy of NPs as drug carriers of antifungal agents. This strategy has also been used to deliver antimicrobial agents for the treatment of *C. neoformans* meningitis. In this case, poly-(n-butyl cyanoacrylate) (PBCA) NPs have been exploited because of their biocompatibility and biodegradability to deliver propolis across the blood–brain barrier (BBB). The invertebrate model *G. mellonella* exhibited normal behavior in toxicity testing, and the treatment with PBCA NPs loaded with propolis increased the percentage of surviving larvae after the infection with *C. neoformans* [191]. Similar results were obtained by developing a PLGA-based nanoformulation in the treatment of *K. pneumoniae* infections: these gentamicin-loaded NPs were able to improve *G. mellonella* survival, thus providing extended prophylactic protection against bacteria [192]. Also, poly-ε-caprolactone (PCL) NPs have been synthesized for drug delivery applications, which were loaded with a new [Ag(I)] coordination complex with antibacterial properties against *Helicobacter pylori*, a Gram-negative bacterium that can colonize the gastric epithelium, promoting gastric pathologies, from peptic ulcer to gastritis and gastric adenocarcinomas. The resulting formulation efficacy has been evaluated in vivo by using *G. mellonella* larvae [193]. Similarly, tobramycin-loaded alginate/chitosan NPs have been developed as an antibacterial formulation against *P. aeruginosa*, and *G. mellonella* in vivo infection experiments showed 90% survival rates after NP injection and low NP toxicity [194]. The *G. mellonella* model is not only involved in the evaluation of antifungal and antibacterial treatments but also is often involved in the evaluation of the efficacy and the biocompatibility of nanoformulations that are applied as anticancer treatments [195,196] and for wound healing [197].

**Table 2 nanomaterials-15-00067-t002:** Administration of functionalized NPs in vivo in *G. mellonella* for toxicity modulation or drug delivery. The in vivo administration of *G. mellonella* using functionalized NPs for toxicity modulation or drug delivery.

	Type	Synthesis Method	Functionalization	NPs’ Average Size (nm)	In Vivo Infection	Toxicity Assessment	Ref.
Inorganic NPs	Ag	Biological| *F. oxysporum*	Phenothiazinium photosensitizers	16–33	- ***	Mortality	[180]
		Biological| *L. acidophilus*	Cinnamaldehyde	9	EAEC	LC_50_–mortality–bacterial load–THC–LDH cytotoxicity assay	[181]
	Ag, CuO, ZnO	Commercial| DENANA NanoBEL- Chemical	Biomolecule coronas	10–50	*A. fumigatus*	Mortality	[182]
	Au nanostar	Chemical	Amikacin	104	*K. pneumoniae*	Mortality	[183]
	Silica	Chemical	Lycopene	Unspecified	- ***	Mortality–cocoon formation	[184]
Polymeric NPs	Alginate	Chemical| external gelation method	Miltefosine	~300	*C. albicans–C. neoformans–C. gattii*	Mortality–fungal burden–histopathological analysis	[185]
		Chemical| external gelation method	Miltefosine	~300	*C. auris*	Mortality–fungal burden–histopathological analysis	[186]
	Lipid-based	Chemical	Itraconazole	216	*S. brasiliensis–C. albicans*	Mortality	[187]
		Chemical| dry liquid film technique	Anidulafungin	~100	*C. albicans*	Mortality	[188]
		Chemical| hot emulsification	*L. sidoides* essential oil	213–445	- *	Mortality	[189]
	PLGA	Chemical| nanoprecipitation	Coumarin 6–pterostilbene	50	*A. brasiliensis*	Mortality	[190]
		Chemical| water-in-oil-in-water	Gentamicin	227	*K. pneumoniae*	Mortality–THC–bacterial load–histopathological analysis	[192]
	PBCA	Chemical| in situ anionic polymerization method	Propolis	~195	*C. neoformans*	Mortality	[191]
	PCL	Chemical| nanoprecipitation	[Ag(I)] complex	155–162	- ***	Mortality	[193]
	Alginate/chitosan	Chemical	Tobramycin–dornase alfa	~500	*P. aeruginosa*	Mortality	[194]
	Acetylated cashew gum NPs	Chemical| nanoprecipitation	Lycopene	160–270	- *	Mortality	[195]
	Cationic nanoemulsion	Chemical	C6 ceramide	~40	- *	Mortality	[196]
	Lecithin/chitosan	Chemical	Melatonin	160–207	- *	Mortality	[197]

* The symbol indicates that *G. mellonella* has been used as an in vivo model for toxicity studies, without in vivo infection. NPs: nanoparticles; Ag: silver; EAEC: enteroaggregative *E. coli*; LC: lethal concentration; THC: total hemocyte counting; LDH: lactate dehydrogenase; CuO: copper oxide; ZnO: zinc oxide; Au: gold; PLGA: poly(lactic-*co*-glycolic) acid; PBCA: polybutylcyanoacrylate PCL: polycaprolactone.

## 5. *G. mellonella* Model for Biocompatibility Assessment of Materials and Nanocomposites

Due to the recent progress in the field of nanotechnology, it is becoming increasingly important to establish the biocompatibility of innovative materials whose use could be transferred to the biomedical field.

Among the most used biopolymers in material sciences, chitosan stands out due to its abundance: it is derived by the deacetylation of chitin, the natural constituent of crustacean exoskeletons and fungal cell walls [198], so it results in a cationic polymer with interesting antimicrobial properties due to electrostatic interactions with the cell wall of Gram-positive bacteria and with the outer cell membrane of Gram-negative bacteria [199]. Chitosan possesses unique properties like biodegradability, biocompatibility, and low toxicity, which make it suitable for potential applications in several different fields, from drug delivery and pharmaceutical formulations [200] to food packaging and preservation [201]. The biocompatibility of chitosan has been already estimated in vivo in rats [202] and zebrafish [203]. Recently, chitosan has been investigated to assess its antifungal activity against multidrug-resistant *C. auris* in the *G. mellonella* model. The in vivo experiments show a reduction in the fungal load and increased survival rates of infected larvae, also confirming the non-toxicity of the biopolymer alone. Also, chitosan shows the capability to induce a stress-like response in *G. mellonella* as a defense mechanism against *C. auris* infections [204]. A chitosan-based nanocomposite material has been developed by encapsulating a natural product, panchovillin, on a chitosan framework by ionotropic gelation. The resulting nanocomposite has been studied in vivo to assess its antimicrobial activity against *Mycobacterium indicus* subsp. pranii using *G. mellonella* larvae as a biocompatibility and infection model [205].

*G. mellonella* has also been involved in investigating the biocompatibility and the antimicrobial efficacy of a self-assembling (naphthalene-2-ly)-acetyl-diphenylalanine-dilysine-OH (NapFFKK-OH) peptide hydrogel for biomedical applications. These in vivo experiments demonstrated a lower toxicity than that one registered in vitro on murine fibroblasts, confirming the importance of an in vivo model that is compatible with the 3R principles of animal testing (Reduction, Refinement, Replacement). Then, the peptide hydrogel’s antimicrobial properties against *S. epidermidis*, *S. aureus*, *E. coli,* and *P. aeruginosa* have been confirmed using *G. mellonella* as an infection model by determining the reduction in the hemolymph bacterial load [206].

In 2020, *G. mellonella* larvae have been proposed for the first time as a burn wound model that could be used to study burn trauma and wound infections without involving traditional mammalian models that entail ethical limitations [207]. Currently, there are several different animal species used in the modeling of wound healing, like rabbit [208], pig [209], and zebrafish [210], but the most used is the murine model because of the cheapness, cost efficiency, and broad knowledge [211,212]. To avoid strict ethical limitations, *G. mellonella* has been proposed as an alternative model of burn trauma and concomitant wound infection with common wound pathogens like *P. aeruginosa*, *S. aureus,* and multi-drug-resistant *A. baumannii* [207]. In lines with this new trend, the antimicrobial efficacy of a peptide hydrogel loaded with polymyxin B has been evaluated for the treatment of *P. aeruginosa* wound infection in the *G. mellonella* burn model, showing a drastic reduction in the mortality percentage from 93% to 13% [213].

*G. mellonella* larvae have been also used as an in vivo model to assess the efficacy of a metal-based coating of glass polyalkenoate cement (GPC) in the inhibition of biofilm formation by MRSA [214]. GPCs have proven to be antibacterial, due to the release of therapeutic ions. In this study, silver- and zinc-containing GPCs have been tested in vivo against MRSA. Once the Zn^2+^ and Ag^+^ ion release from GPCs was determined by Atomic Absorption Spectroscopy (AAS), the elutes were tested in vivo by infecting *G. mellonella* larvae, and subsequently injecting them with the scaffold elutes. The results showed that the levels of Zn^2+^ and Ag^+^ released from the GPCs are at a concentration high enough to reduce the viability of MRSA in vitro. In vivo experiments confirmed these results, that is, the antimicrobial function of GPC elutes proportionally correlated with the concentration of metal ions [214]. *G. mellonella* has been also involved in the evaluation of biofilm-associated infections on stainless and titanium implants [215]. First of all, the biocompatibility was evaluated in vivo with sterile implants, and they showed no significant adverse effects over the observation period of 5 days. Then, stainless and titanium implants contaminated with *S. aureus* were implanted in vivo to mimic the biofilm-associated infection, leading to a significative reduction in the larval survival. The in vivo model also confirmed the efficacy of gentamicin in the treatment of planktonic infection with *S. aureus*, and its inefficacy in the case of MRSA biofilm-associated infection of the implants [215]. As a result, *G. mellonella* proved itself to be an adequate non-mammalian model to determine the antibacterial efficacy of biofilm-associated infections. Recently, *G. mellonella* has been used as a burn wound infection model to assess the safety and the efficacy of a bio-based formulation made with bacterial cellulose and a mixture of neem and hypericum oil against *P. aeruginosa* and *S. aureus* skin lesion infections [216].

## 6. Standardization in Larvae Rearing and Experimental Protocols

Despite the numerous studies reported in this review, much work remains to be performed to establish *G. mellonella* as a standardized model for animal experimentation. First, it has been shown that the diet provided to larvae can significantly impact their health, thus influencing the outcomes of infection experiments [34]. Temperature and rearing conditions are also crucial in infection studies [75]. Likewise, the injection methods for pathogens must be standardized. For instance, the inoculum solution can contain various buffers (such as Phosphate-Buffered Saline—PBS) or aqueous solutions with salts like magnesium sulfate (MgSO_4_), and different protocols can be used to prepare the inoculum [217,218]. The most common infection method involves injecting the pathogen into the hemocoel through the left proleg using a 0.75 mm diameter needle [40,219]. It is essential to perform specific control experiments, such as the mock inoculation of larvae with only the solution used to resuspend the microorganisms, to ensure that the solution does not affect larval viability. Many researchers also recommend mock inoculation to ensure that handling and inoculation procedures do not adversely affect larval health [219]. Careful handling is important, as rough handling can affect survival rates and induce the expression of stress proteins [75]. Furthermore, the larvae have a complex life cycle [34], so age and weight should be standardized when selecting larvae for experiments.

Another critical aspect is the identification of key parameters to monitor during infection. Typically, inoculated larvae begin to darken (turn brown/black) shortly after inoculation, depending on the virulence of the pathogen. As previously described, this color change is associated with the onset of myelination. Other parameters to be assessed include larval mortality, microbial counts after infection, and changes in the expression of antimicrobial peptides. The expression of antimicrobial peptides can be measured using techniques such as RT-qPCR [14,89,93] or through SDS-PAGE and autoradiography experiments [89].

A significant advancement in the standardization of *G. mellonella* as a model organism is the availability of commercial products. One such example is TruLarv [220], a standardized larvae model developed by BioSystems Technology. This model offers several advantages over traditional methods, including enhanced consistency in experimental results. The larvae are reared without antibiotics or hormones, standardized for age and weight, genetically homogeneous (from a sequenced and inbred breeding colony), and feature a decontaminated cuticle.

To further advance the use of *G. mellonella* as a reliable animal model, it is crucial that, in the coming years, competent authorities issue guidelines for its use, adhering to the ethical principle that the use of mammals in biomedical research should be minimized.

## 7. Conclusions and Perspectives

The health emergency caused by bacteria resistant to traditional antibiotics necessitates the rapid development of new therapies and health technologies. Among these innovations are nanoparticles and innovative materials. Different physico-chemical aspects, such as size, shape, charge, and chemical composition, can influence in vivo biodistribution and efficiency of nanoparticles and nanomaterials. Furthermore, the widespread use of nanoparticles and nanomaterials in biomedical and industrial fields, such as food packaging, tissue engineering, and wound healing, leads to the need to evaluate their nanotoxicity with increasingly reliable, simple, and economical techniques. In this context, the use of *G. mellonella* as an economic and easily available animal model can promote in vivo investigations alternative to in vitro tests and to traditional mammalian and non-mammalian animal models without ethical limitations.

Despite the advantages of this successful animal model, there are several shortcomings that need to be addressed. Firstly, *G. mellonella* larvae are often purchased in fishing shops or pet stores as larvae are used as diet supplements for amphibians and reptiles. This fact sometimes results in poor reproducibility of the experiments conducted by different research groups due to the genetic variability of the animals. From this point of view, the use of pure lines of *G. mellonella* is necessary. Secondly, the growth and feeding conditions of the larvae can influence their health status, their innate immune response, and, consequently, their mortality in experiments. For this reason, the standardization of feeds and growth protocols is crucial. Finally, a further step could be the development of humanized transgenic larvae.

All aspects listed above are prerequisites for regulatory authorities to produce policies to approve *G. mellonella* as an animal model according to the 3R principles of animal testing. Advances in the standardization of the use of *G. mellonella* as an animal model, and related cost reductions, could be one of the elements that will boost research on antimicrobial drugs, including nanotechnological formulations such as nanoparticles and nanostructured materials.

## Figures and Tables

**Figure 1 nanomaterials-15-00067-f001:**
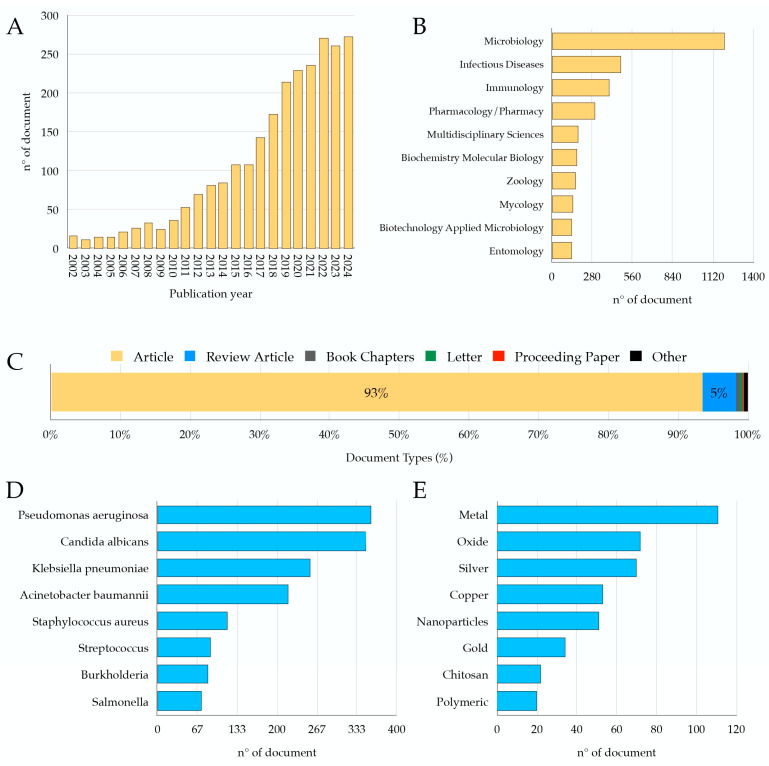
Indexed documents in WOS retrieved using the keywords “*Galleria mellonella* infection”. (**A**) Number of papers published in the years 2002 to 2024. (**B**) Breakdown of identified documents according to the category in which they are classified in the WOS database. (**C**) Types of articles published in the years 2002 to 2024. (**D**) Pathogenic microbes frequently used in infection experiments with *Galleria mellonella*. (**E**) Materials and elements most referenced in studies involving *Galleria mellonella*.

**Figure 2 nanomaterials-15-00067-f002:**
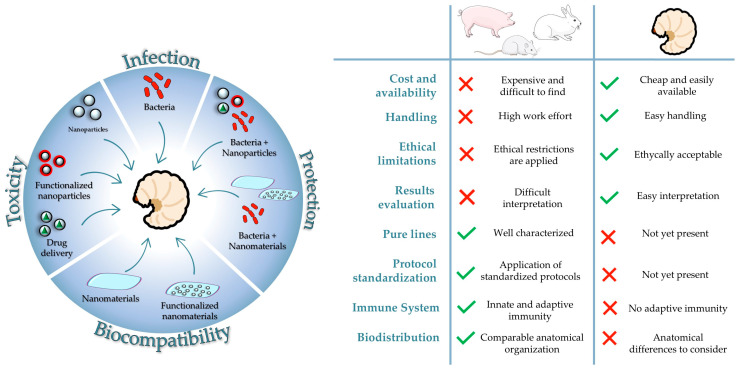
On the left, the schematic representation of *G. mellonella* as a reliable animal model for experiments of infections, the evaluation of toxicity and biocompatibility of nanomaterials, and the evaluation of the antimicrobial efficacy of nanotechnologies against pathogens. On the right, the summary of the main advantages and disadvantages of using *Galleria mellonella* when compared to mammalian models.

**Figure 3 nanomaterials-15-00067-f003:**
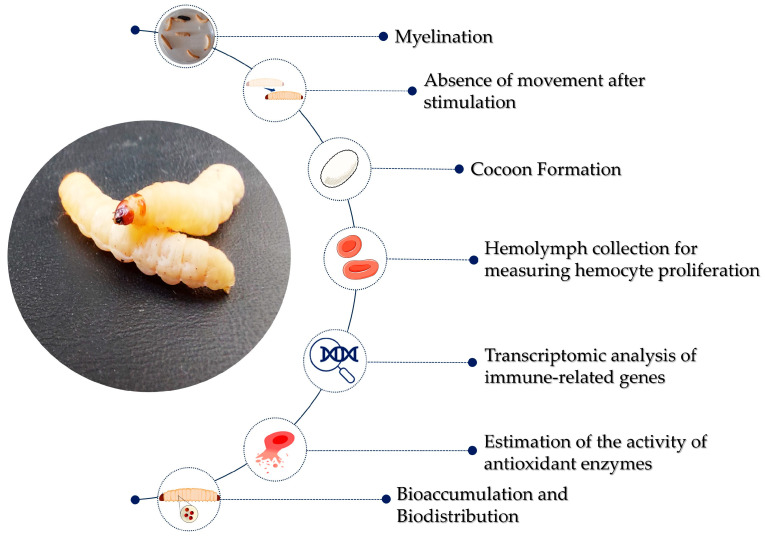
A summary of the main strategies available for the evaluation of the toxicity of chemicals and nanomaterials in the *Galleria mellonella* in vivo model.

**Figure 4 nanomaterials-15-00067-f004:**
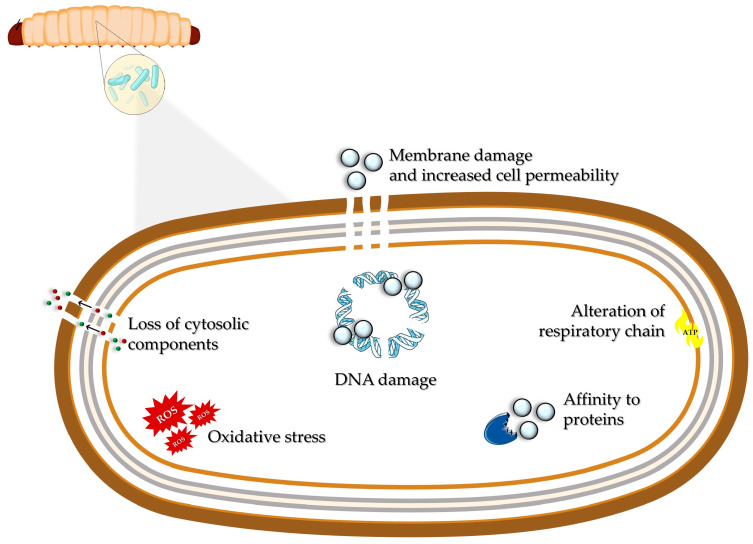
The schematic representation of the mechanisms of action of nanoparticles when administered as antimicrobial agents in *Galleria mellonella*-infected larvae.

## Data Availability

All the data used for this review are available in the cited documents.

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
