# Peer review of "Galleria mellonella (Greater Wax Moth) as a Reliable Animal Model to Study the Efficacy of Nanomaterials in Fighting Pathogens"

_nanomaterials, 2025, doi:10.3390/nano15010067_

Round 1
Reviewer 1 Report
Comments and Suggestions for Authors
This study introduced the using Galleria mellonella model to learn the efficacy and safety of NPS/nanomaterials. It could provide the information for investigaing the in vivo biological behavior of the formulated NPS. The following issues need to be suggested.
1. The comparation of other methods for evaluating the toxicity and efficacy were suggested to be added in this work.
2. The statistical data need to be added to show the reliability of the animal model.
3. For each section, It is suggested to add some figures.
4. The recent published paper need to be also involved in Figure 1.
5. Please give some details about the regulation of using this model for in vivo study.
Author Response
We thank the reviewer for his comments that helped improve the manuscript. The responses to the comments are given below.
This study introduced the using Galleria mellonella model to learn the efficacy and safety of NPS/nanomaterials. It could provide the information for investigaing the in vivo biological behavior of the formulated NPS. The following issues need to be suggested.
Comment 1: The comparation of other methods for evaluating the toxicity and efficacy were suggested to be added in this work.
Response 1: We agree with the reviewer’s suggestion. We have added a discussion on in vitro cell models and mammalian models (lines 336-377). Additionally, we have included a new figure (Figure 2) representing a graphical summary of the advantages and disadvantages of mammalian models and G. mellonella.
Comment 2: The statistical data need to be added to show the reliability of the animal model.
Response 2: As this is a review, we do not believe it is necessary to include statistical data. In fact, the usefulness of the model can be directly inferred from several studies referenced in the manuscript.
Comment 3: For each section, it is suggested to add some figures.
Response 3: Thank you for your suggestion. We added Figure 2, Figure 3, and Figure 4 to the manuscript.
Comment 4: The recent published paper need to be also involved in Figure 1.
Response 4: We agree with the reviewer’s suggestion. We have revised the figure caption and updated the figure as requested.
Comment 5: Please give some details about the regulation of using this model for in vivo study.
Response 5: So far, there are no guidelines regulating the use of Galleria mellonella larvae as an in vivo model. In the member states of the European Union, Directive 2010/63/EU is valid, which regulates animal experimentation, introducing the principles of the 3Rs (Replacement, Reduction, Refinement). This Directive, however, does not concern the use of Galleria mellonella. The applicability of Galleria mellonella in the absence of ethical restrictions is better discussed in lines 91-93.
Reviewer 2 Report
Comments and Suggestions for Authors
Comments
The topic of the paper is very interesting, and it includes an extensive literature review on the use of Galleria mellonella as an alternative animal model not only to study the toxicity of nanoparticles and nanomaterials (mainly), but also new antimicrobial strategies can also use this animal model. The paper is very well presented and holds the reader's attention from beginning to end. However, some minor points should be addressed.
1) Very big title. My proposal: Galleria mellonella as a reliable animal model to study the toxic properties of antimicrobial nanoparticles.
2) It would be good to include a graphic mechanism of action of nanomaterials on the death or survival of Galleria mellonella larvae.
3) I don't find the graphical summary for which you are grateful.
Author Response
We thank the reviewer for his comments that helped improve the manuscript. The responses to the comments are given below.
The topic of the paper is very interesting, and it includes an extensive literature review on the use of Galleria mellonella as an alternative animal model not only to study the toxicity of nanoparticles and nanomaterials (mainly), but also new antimicrobial strategies can also use this animal model. The paper is very well presented and holds the reader's attention from beginning to end. However, some minor points should be addressed.
Comment 1: Very big title. My proposal: Galleria mellonella as a reliable animal model to study the toxic properties of antimicrobial nanoparticles
Response 1: We agree with this suggestion. We modified the title as reported in the manuscript.
Comment 2: It would be good to include a graphic mechanism of action of nanomaterials on the death or survival of Galleria mellonella larvae.
Response 2: Thank you for your suggestion. We add Figure 4 to provide a general overview about the mechanism of action of nanomaterials used as antimicrobial agents in Galleria mellonella bacterial infections.
Comment 3: I don’t find the graphical summary for which you are grateful.
Response 3: We are so sorry for the inconvenience. We uploaded the graphical summary in Figure 2.
Reviewer 3 Report
Comments and Suggestions for Authors
The manuscript titled “Fighting pathogens with nanotechnology: Galleria mellonella to study the biological properties of antimicrobial nanoparticles in a reliable animal model” is a well-structured and informative review. It comprehensively summarizes relevant studies, offering significant insights into G. mellonella’s immune system and its advantages in nanotoxicity assessment over traditional animal models. The manuscript is a valuable contribution to the field and addresses a relevant topic. However, there are areas where improvements could enhance its impact.
1. To enhance the reader's understanding and engagement, consider adding a schematic illustration that provides an overview of the topic. This could include the immune responses of G. mellonella and their relevance to nanoparticle interactions.
2. It would benefit from a critical discussion of the limitations of using G. mellonella for nanotoxicity studies. For example, highlight potential discrepancies between nanoparticle effects observed in insects versus mammals, particularly concerning metabolism, biodistribution, and immune system differences.
3. Include a more critical analysis of whether the results obtained using G. mellonella are reliable indicators of mammalian efficacy. Discuss specific challenges or conditions where this model might fail to accurately predict mammalian responses. If available, include a direct comparison of nanotoxicity or antibacterial efficacy results between G. mellonella and mammalian models. Highlighting such findings would strengthen the argument for the model's relevance and applicability.
4. The manuscript provides extensive details about nanoparticle terminology. Consider condensing this section to focus more on toxicity and biological interactions, which are central to the review.
Author Response
We thank the reviewer for his comments that helped improve the manuscript. The responses to the comments are given below.
The manuscript titled “Fighting pathogens with nanotechnology: Galleria mellonella to study the biological properties of antimicrobial nanoparticles in a reliable animal model” is a well-structured and informative review. It comprehensively summarizes relevant studies, offering significant insights into G. mellonella’s immune system and its advantages in nanotoxicity assessment over traditional animal models. The manuscript is a valuable contribution to the field and addresses a relevant topic. However, there are areas where improvements could enhance its impact.
Comment 1: To enhance the reader’s understanding and engagement, consider adding a schematic illustration that provides an overview of the topic. This could include the immune response of G. mellonella and their relevance to nanoparticle interactions.
Response 1: Thank you for your suggestion. We added Figure 4 with the intent to show the mechanism of action of nanoparticles if administered as antimicrobial agents in G. mellonella infected larvae.
Comment 2: It would benefit from a critical discussion of the limitations of using G. mellonella for nanotoxicity studies. For example, highlight potential discrepancies between nanoparticles effects observed in insects versus mammals, particularly concerning metabolism, biodistribution, and immune system differences.
Response 2: We agree with the suggestion of the reviewer. We add Figure 2 as a schematic summary of advantages and disadvantages of G. mellonella versus mammals.
Comment 3: Include a more critical analysis of whether the results obtained using G. mellonella are reliable indicators of mammalian efficacy. Discuss specific challenges or conditions where this model might fail to accurately predict mammalian responses. If available, include a direct comparison of nanotoxicity or antibacterial efficacy results between G. mellonella and mammalian models. Highlighting such findings would strengthen the argument for the model’s relevance and applicability.
Response 3: Thank you for this suggestion. We better discussed the adoption of G. mellonella as in vivo model in comparison to traditional mammalian models in lines 336-377.
Comment 4: The manuscript provides extensive details about nanoparticle terminology. Consider condensing this section to focus more on toxicity and biological interactions. Which are central to the review.
Response 4: As this review investigates the applicability of Galleria mellonella in the field of nanotechnology as a model for the evaluation of in vivo antimicrobial efficacy, we believe it is important to ensure accuracy in the analysis of the fabrication and characterization of nanomaterials, together with the biological component.
Round 2
Reviewer 1 Report
Comments and Suggestions for Authors
The MS is suggested to be published
Author Response
We sincerely thank the reviewer for their time and effort in reviewing our work